# Evidence for influenza and RSV interaction from 10 years of enhanced surveillance in Nha Trang, Vietnam, a modelling study

**Naomi R. Waterlow**[1]*, **Michiko Toizumi**[2], **Edwin van Leeuwen**[1,3], **Hien-Anh Thi Nguyen**[4], **Lay Myint-Yoshida**[2°], **Rosalind M. Eggo**[1°], **Stefan Flasche**[1°]

**1** Centre for Mathematical Modelling of Infectious Disease, London School of Hygiene and Tropical Medicine, London, United Kingdom, **2** Department of Pediatric Infectious Diseases, Institute of Tropical Medicine, Nagasaki University, Nagasaki, Japan, **3** Statistics, Modelling and Economics Department, UKHSA, London, United Kingdom, **4** National Institute of Hygiene and Epidemiology, Hanoi, Vietnam

° These authors contributed equally to this work.
* naomi.waterlow1@lshtm.ac.uk

**Data Availability Statement:** The code and summarised data underlying this project are

## Abstract

Influenza and Respiratory Syncytial Virus (RSV) interact within their host posing the concern for impacts on heterologous viruses following vaccination. We aimed to estimate the population level impact of their interaction. We developed a dynamic age-stratified two-pathogen mathematical model that includes pathogen interaction through competition for infection and enhanced severity of dual infections. We used parallel tempering to fit its parameters to 11 years of enhanced hospital-based surveillance for acute respiratory illnesses (ARI) in children under 5 years old in Nha Trang, Vietnam. The data supported either a 41% (95% CrI: 36–54) reduction in susceptibility following infection and for 10.0 days (95%CrI 7.1–12.8) thereafter, or no change in susceptibility following infection. We estimate that co-infection increased the probability for an infection in <2y old children to be reported 7.2 fold (95% CrI 5.0–11.4); or 16.6 fold (95%CrI 14.5–18.4) in the moderate or low interaction scenarios. Absence of either pathogen was not to the detriment of the other. We find stronger evidence for severity enhancing than for acquisition limiting interaction. In this setting vaccination against either pathogen is unlikely to have a major detrimental effect on the burden of disease caused by the other.

## Author summary

Influenza and Respiratory Syncytial Virus (RSV) cause large burdens of disease. Instead of acting independently, there may be short term cross-protection between them. The evidence of this to date comes from ecological studies which are unable to test the mechanism, or biological studies that are unable to determine the population level impacts of such cross-protection. We create a mathematical model that simulates the circulation of these two viruses, and allows for cross-protection between them. We then fit this model to hospital reported cases of confirmed infection from Nha Trang, Vietnam in order to estimate whether any cross-protection exists in this setting. We show that there are two

available at https://github.com/NaomiWaterlow/NhaTrang_flu_rsv_interaction.

**Funding:** NRW was supported by the Medical Research Council (grant number MR/N013638/1). EvL and RME declare funding from the National Institute for Health Research (NIHR) Health Protection Research Unit (HPRU) in Modelling and Health Economics, a partnership between PHE, Imperial College London, and LSHTM (grant number NIHR200908). EvL was supported by the European Union's Horizon 2020 research and innovation programme - project EpiPose (101003688). SF is funded through a Sir Henry Dale Fellowship jointly funded by the Wellcome Trust and the Royal Society (grant number 208812/Z/17/Z). RME acknowledges an HDR UK Innovation Fellowship (grant: MR/S003975/1), MRC (grant: MC_PC 19065), and NIHR (grant: NIHR200908) for the Health Protection Research Unit in Modelling and Economics at LSHTM. The views expressed in this publication are those of the author(s) and not necessarily those of the funders. This ARI surveillance study in Nha Trang was supported by Japan Program for Infectious Diseases Research and Infrastructure, Japan Agency for Medical Research and Development (AMED) under Grant Number JP21wm0125006. The funders had no role in the study design, data collection and analysis, decision to publish, or preparation of the manuscript.

**Competing interests:** The authors have declared that no competing interests exist.

possibilities—either no interaction or moderate interaction that can result in the observed circulation patterns. However, we further show that co-infection results in an increased reporting rate, presumably due to increased severity.

## Introduction

Influenza and Respiratory Syncytial Virus (RSV) have large health and economic impacts globally, particularly in young children where they cause 870 000 [1] and 3.2 million [2] hospitalisations in <5 year olds per year respectively. While pediatric influenza vaccines are licensed for use in some countries, global uptake is poor and efficacy depending on the match to the circulating strains. RSV vaccines are in development, with close to 20 vaccine candidates being evaluated in pre-licensure trials [3].

The impact of vaccination may be enhanced if concurrent co-infections increase the propensity of severe disease beyond that of either pathogen [4]. However, the impact of vaccination may be lessened if vaccination reduces competitive pressure between influenza and RSV and thus leads to increased circulation of the other pathogen. Such competitive pressure has been observed in the form of cross protection in mouse studies that showed e.g. a protective effect of live attenuated influenza vaccine administration on RSV replication [5] and influenza infection on RSV severity [6], mediated by the innate immune system. Population level evidence for the effect of cross-protection on influenza and RSV epidemiology, however, is largely of observational nature: a lack of coincidence in peak timings [7,8], changes in RSV peak timing following unusual influenza seasons [9–14] and alternating infection patterns [15].

Cross-protection following a primary infection could occur through a variety of mechanisms including viral competition for resources in the host [16], the activation of the innate immune system such as through toll-like receptors (TLRs) 3 and 7 [17,18] or short term immune memory through surviving cells in an antiviral state (e.g. epithelial cells following influenza infection [19]). These interactions could result in a reduction in subsequent infection with the other viruses, and estimates of the duration of cross-protection and its biological pathway vary. Experimental infection of ferrets estimated less than 2 weeks protection between influenza A and B viruses [20], yet cells forming the respiratory epithelium can survive in a state of heightened antiviral activation for 3 to 12 weeks after influenza A infection, with waning of the conferred protection observable at 6 weeks [19].

In Nha Trang, Vietnam, for more than 10 years children admitted to the single public hospital with acute respiratory illness have been tested for presence of influenza and RSV infection as part of an enhanced surveillance. RSV circulation is highly seasonal, and influenza circulation varies year on year, thus giving a unique opportunity to systematically investigate evidence for their interaction at population level. We use this data in combination with a dynamic transmission model to estimate the strength of cross-protection and the effects of co-infections on disease severity (defined as the proportion of infections that require hospital attendance).

## Methods

### Ethics statement

This study was approved by the Institutional Review Boards of the London School of Hygiene & Tropical Medicine (16166 /RR/12988) and the National Institute of Hygiene and Epidemiology in Vietnam (VN01057-28/2015).

## Study population

We used data from a hospital-based enhanced surveillance study of children with respiratory disease, as previously described [21,22]. In brief, children younger than 5 years old who resided in 16 out of the 27 communes of Nha Trang (resident population of 210'739) and attended he paediatric ward in Khanh Hoa General Hospital (KHGH) in Nha Trang, central Vietnam, with Acute Respiratory Infection (ARI) were enrolled and offered a suite of additional diagnostics. ARI was defined as Cough and / or difficulty breathing. Khanh Hoa hospital is a tertiary care facility and is the only public hospital for the catchment area of the study. More than 94% of all pediatric ARI admissions are typically enrolled. Upon admission, Nasopharyngeal (NP) samples were taken from patients, nucleic acid was extracted and multiplex-PCR assays were performed in order to detect infection with up to 13 respiratory viruses, including influenza A and RSV. Positive samples underwent a second, confirmatory PCR test and only individuals who tested positive in both PCRs were included. We use aggregate weekly data, from 5th February 2007 until 4th December 2017. We assumed that an ARI episode, for which Influenza or RSV were detected from the nasopharynx on admission, was caused by the respective pathogen. The dataset excludes neonatal cases under 28 days old.

To inform transmission pathways in the population we used age specific contact-patterns, based on a contact study in the same area, conducted in 2010[23]. In total 2002 Nha Trang residents of all ages participated in the study. A contact was defined as either skin-to-skin contact or a two-way conversation.

## Data analysis

We calculated the correlation between all reported influenza and RSV cases each week using a Pearson's Correlation test.

Assuming no interaction (in susceptibility to or severity of dual infections), we calculated the required annual RSV infection attack rate in order to observe the reported number of dual infections (Eqs 1–3). We estimate the RSV attack rate rather than the influenza attack rate, as RSV is more consistent year on year (see section 1 in S1 Text for influenza equivalent). Using a negative binomial likelihood with Brent optimization we estimated the RSV reporting rate that would correspond to the maximum likelihood of observing the reported weekly number of dual infections. We then used this estimate of the reporting rate to calculate the annual RSV population attack rate required in order to observe this many coinfected ARI admissions. The credible intervals for the attack rate were calculated using the Hessian matrix from the optimisation. If the estimated attack rate is high, this may suggest that influenza and RSV co-infection increase severity (and hence reporting). If the estimated attack rate is low, this could suggest that co-infection is less likely that at random due to competition between the viruses.

$$I_{Dual} \simeq I_{Influenza} * P_{RSV} \tag{1}$$

$$P_{RSV} \simeq I_{RSV} * 1/\gamma_{RSV}/\upsilon_{RSV} \tag{2}$$

$$AR_{RSV} \simeq I_{RSV}/\upsilon_{RSV} \tag{3}$$

With parameters: Incidence of reported cases (I), Prevalence of Infection (P), Duration of Infection ($1/\gamma_{RSV}$, 9 days—see Table 1), attack rate (AR) and estimated reporting rate ($\upsilon$).

**Table 1. Parameter definitions, values and priors.**

| Parameter | Symbol | Value | Prior | Source |
|---|---|---|---|---|
| Transmission rate INF | $\beta_{INF}$ | Fitted | Based on $R_{0,INF}$ | See section 4 in S1 Text for calculations |
| Basic Reproduction Rate Influenza | $R_{0,INF}$ | Fitted | Between 1 and 8 | [26] |
| Transmission rate RSV | $\beta_{RSV}$ | Fitted | Based on $R_{0,RSV}$ | See section 4 in S1 Text for calculations |
| Basic Reproduction Rate RSV for strain y | $R_{0,RSV,y}$ | Fitted | Between 1 and 8 | [26] |
| RSV age group susceptibility (0–1, 2–4, 5–15, 16–64,65+) | $\tau_i$ | Fixed | 1, 0.75, 0.65, 0.65, 0.65 | Based on Henderson et al (1979) [27], see section 5 in S1 Text |
| Infectious period Influenza | $1/\gamma_{INF}$ | 3.8 days | - | Cauchemez et al (2004) [28] Range from published papers: 1–4.5 days [28–31] |
| Infectious period RSV | $1/\gamma_{RSV}$ | 9 days | - | Weber et al (2001) [25] Range from published papers 6.7–12 days [25,32,33] |
| Strength of cross-protection | $\sigma$ | Fitted | 0–1 | Assuming competitive [34] |
| Duration of cross-protection | $1/\rho$ | Fitted | 0—Inf days | |
| Proportion of each age group infected with Influenza, at the start of the season | $\delta_{INF,s}$ | Fitted | 0–1 | |
| Proportion of each age group infected with RSV, at the start of the season | $\delta_{RSV,s}$ | Fitted | 0–1 | |
| Proportion susceptible to influenza at the start of the season | $\eta_s$ | Fitted | 0–1 | Exponential function, see section 6 in S1 Text for details |
| Influenza proportion reported in ages 0–1 | $\kappa_{INF}$ | Fitted | 0–0.4 | |
| Influenza multiplier for proportion reported ages 2–4 vs 0–1 | $\kappa_{INF,m}$ | Fitted | 0–5 | |
| RSV proportion reported in age group i | $\kappa_{RSV,i}$ | Fitted | 0–0.4 | No additional severity from dual infection [35] |
| RSV ON-1 reporting multiplier | $\kappa_{RSV,2012}$ | Fitted | 1–5 | ON-1 clinically more severe [24] |
| Dual infection multiplier on RSV proportion reporting | $\kappa_{Dual}$ | Fitted | 1—Inf | Based on analysis of expected RSV Attack Rate above. |
| Overdispersion parameter | $k$ | Fitted | 0-Inf | |

## Model

We created an age-structured deterministic transmission model for influenza and RSV, allowing for short-term cross-protection. Individuals are either Susceptible (S), Infectious (I), cross-Protected (P) and Recovered (R) for influenza (INF) and (RSV).

Susceptibles become infected at force of infections $\lambda_{INF}$ and $\lambda_{RSV}$, and move into the I states. They then remain infectious for $1/\gamma_{INF}$ and $1/\gamma_{RSV}$ days and during the infectious period and $1/\rho$ days thereafter they are cross-protected and thus their propensity for heterologous infection is reduced by factor $\sigma$, the strength of cross-protection. There is no difference in inherent susceptibility by age for influenza, but there is reduced susceptibility to RSV in older age groups, determined by parameter $\tau_i$.

The force of infection includes age-specific contact rates derived from a local contact survey [23]. Modelled age-groups are: 0–1 years (infants), 2–4 years (pre-school), 5–15 years (school), 16–64 years (adults) and 65 + (older adults). Infection reporting rates vary by age-group and virus, and for RSV reporting rates are increased by a multiplier from 2012 onwards, due to the circulation of a new genotype that has increased the average severity of infection and thus the proportion of reported infections (ON-1) [24]. There is also a multiplier on the RSV reporting rates for dual infections, allowing them to be reported more frequently, for example because of increased propensity for respiratory disease that would require healthcare seeking (as observed in adults [4]). Model equations are shown in section 3 in S1 Text.

We model the persisting immunity to influenza at the start of each season individually, with an initial proportion susceptible. This allows for a different immunity profile at the start of each season, without needing to account for waning of immunity, dynamics across virus

subtypes and seasonality specifically. [25]. Due to infections in previous influenza seasons and potential vaccination, susceptibility to influenza is assumed to decline exponentially at rate $\eta$ with age (see section 6 in S1 Text). This is modelled as non-leaky cross-protection, due to the combination of different exposures. For RSV we assume the only immunity at the beginning of the season is the age-specific reduction in susceptibility (leaky immunity), as immunity to reinfection typically lasts less than a year. Due to the short-time period modelled (max 66 weeks), we do not include births, deaths or ageing, but instead hold the age-group specific population sizes constant at the levels of 2010. Parameter definitions and values are shown in Table 1.

To capture the annual influenza and RSV epidemics despite regular changes, particularly in the timing of influenza circulation, we defined the annual start of the season as the minimum number of combined RSV and Influenza cases (on a 4 week rolling mean) between the 1st of November and the 1st of May each season. If one or more weeks had the same rolling average, we took the first occurrence within the time window.

## Likelihood

We fitted the model to the age-stratified weekly number of ARI cases with nasopharyngeal carriage of either influenza or RSV using a negative binomial likelihood. To fit the allocation of those cases into influenza, RSV or dual infections we added a multinomial component to the likelihood, resulting in an overall log likelihood of:

$$LL(\theta|x) = \sum_{j=1}^{N=2} \sum_{i=1}^{n} (NB(\mu_{i,j}, k) + MN(pRSV_{i,j}, pFlu_{i,j}, pDual_{i,j})) \tag{4}$$

Where $x$ are the reported infections, $\theta$ are the parameters, $j$ are the two age groups 0–1 and 2–4, and $i$ are the weeks, with n being the total number of weeks. NB is the likelihood of the observed number of cases being a random draw from a negative binomial distribution with the total number of modelled infections as the mean, $\mu$, and the fitted overdispersion parameter, $k$. MN is the multinomial likelihood, with $pRSV$, $pFlu$ and $pDual$ being the respective probabilities of the infection with influenza, RSV or both, calculated from the ratio of model reported cases.

## Inference

We used parallel tempering to fit the model. This method involves running multiple markov chains simultaneously, with different 'temperatures' that place a weighting on the likelihood. Swaps between the chains are then proposed every x (in this case 5) iterations, and accepted with acceptance ratio:

$$R = e^{((LL(i)-LL(j))*(\tau_j - \tau_i))}$$

Where

$$\tau_i = \frac{1}{T_i}$$

For full details of the method see Vousden *et al* (2016)[36] We ran the parallel tempering algorithm with 12 chains and 450,000 iterations. The initial 250,000 iterations were discarded as burn-in. Accepted samples from the first chain were then thinned to 1 in 10 for analysis, resulting in a final sample size of 20'000.

### Single pathogen simulations

To assess the maximal indirect heterologous effect an intervention against either pathogen (e.g. widespread vaccination)could have in this environment, we assumed that complete absence of one of the viruses. We then calculated the relative change in the number of cases in the pathogen not targeted by the intervention. Estimates are based on model simulations using 1000 posterior parameter samples of the fitted model.

### Sensitivity analyses

We assessed the sensitivity of our estimates of the strength of interaction of influenza and RSV to the prior on the interaction parameter and to the reporting rates of dual infections. We reran the model with a prior on the interaction parameter for strong interaction, using a normal distribution with mean 0.8 and standard deviation of 0.15. In addition, we ran a version of the model which did not allow an increased reporting rate for dual infections, as it has been reported that in this setting there is no increased severity of dual infections among hospitalised children [35]. Instead, it was assumed that the reporting rate for dual infections was the same as that for RSV-only infections.

### Software

All analysis except for the fitting was done in R version 4.0.0. The fitting was done on R 3.4 on AWS ec2 machines. Code is available at https://github.com/NaomiWaterlow/NhaTrang_flu_rsv_interaction

## Results

### Descriptive analysis

A total of 788 influenza and 1687 RSV hospitalised paediatric ARI cases were reported between 5th February 2007 and 4th December 2017; 78 (9% of influenza cases and 4% of RSV cases) of these were dual infections (Fig 1A and 1B). The mean age of at admission was 22 months and 16 months for influenza and RSV cases respectively. RSV notifications showed strong consistent seasonality across years, peaking usually in the 34th week of the year, whereas influenza showed less seasonality, but typically occurs after Tết Nguyên Đán holidays and before the RSV epidemic (Fig 1C).

There was a small, not statistically significant, negative correlation between weekly influenza and RSV incidence, with the Pearson correlation coefficient -0.074 (95% CrI:-0.160 to 0.009). (Figure A in S1 Text)

We estimated that in order to observe the weekly reported number of dual infections when assuming independence of influenza and RSV infection, the annual RSV attack rate needed to be 720% (95% CrI: 560–1000) in ages 0-1y and 430% (95% CrI: 270–980) in ages 2-4y. The high attack rate suggests that in fact influenza and RSV infections are not independent but that co-infection is likely to substantially enhance the propensity for hospitalisation with ARI in this setting.

### Model inference

The model was able to fit the case data for influenza and RSV well (Fig 2, see section 7 in S1 Text for further fitting and convergence details). The posterior estimates for the relative reduction in heterologous acquisition rates during and following Influenza or RSV infection was bimodal, with one mode at 0.004 (95%CrI 0.000–0.046), indicating hardly any competition for infection, and one mode at 0.41 (95%CrI 0.36–0.54), indicating moderate competition,

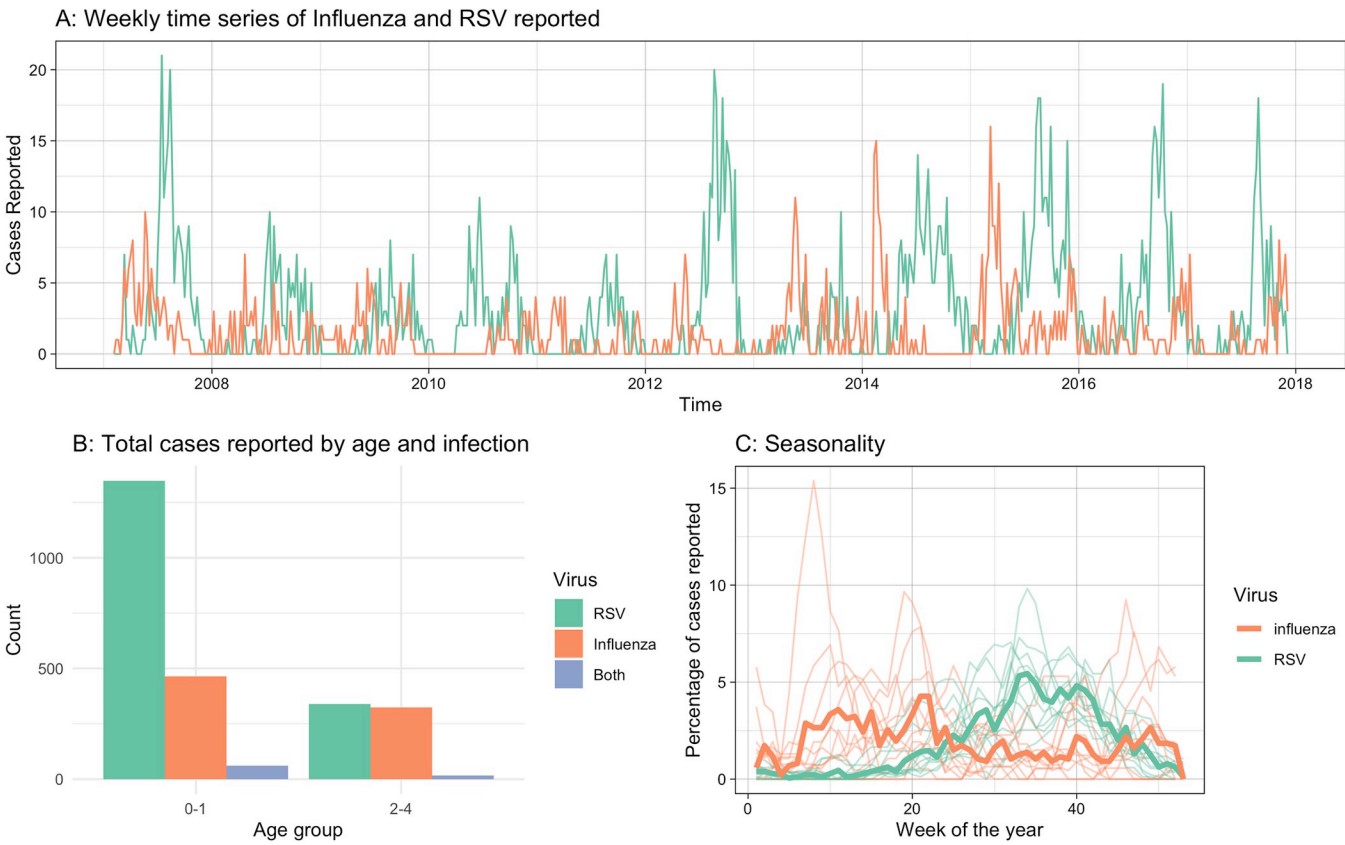

**Fig 1. Data.** A) Weekly reported infections of children under 5 years old infected with influenza and RSV over the study period. B) Total number of cases reported over the entire study period by age group and virus. C) Percentage of reported cases by week of the year for RSV and Influenza. The thick lines show the combined total reported across all years, the semi-transparent lines show the 4-week moving average per year.

assuming a cut-off between modes at 0.2 (Fig 3). The posterior for the duration of interaction also had multiple modes, with the mode corresponding to moderate competition at 10.0 days (95%CrI 7.1–12.8 days).

The main differences between modes for other parameters were in the detection rate of influenza, which ranged from 13 to 21% of infections reported (section 8 in S1 Text) and the increased reporting for dual infections. We estimate that in the moderate competition mode the observation of influenza and RSV coinfection among ARI cases was 8.2 (95%CrI 6.9–9.9) times more likely than would have been expected by chance in ages 2–4 and 16.6 (95%CrI 14.5–18.4) in ages 0–1. This compares to the no competition scenario where the observation of influenza and RSV coinfection among ARI cases was 3.6 (95%CrI 2.5–5.8) and 7.2 (95%CrI 5.0–11.4) times more likely than would have been expected by chance in ages 2–4 and 0–1 respectively.

To assess the relevance of RSV and influenza interaction on population level in this setting we simulated single pathogen versions of the parameterised model. In the case of no competition, absence of influenza (e.g. through widespread vaccination) reduced RSV hospitalisations by 4.1% (95%CrI 3.3–7.1%) due to a lack of co-infections with higher propensity for severe disease and absence of RSV reduced influenza hospitalisations by 7.2% (95%CrI 4.4–7.2%) in the study period. In the moderate competition mode, absence of influenza reduced RSV hospitalisations by 5.7% (95%CrI 4.9–6.5%). In the absence of RSV 1.8% (95%CrI -0.7–7.2%) more hospitalised cases for influenza occurred (Fig 4).

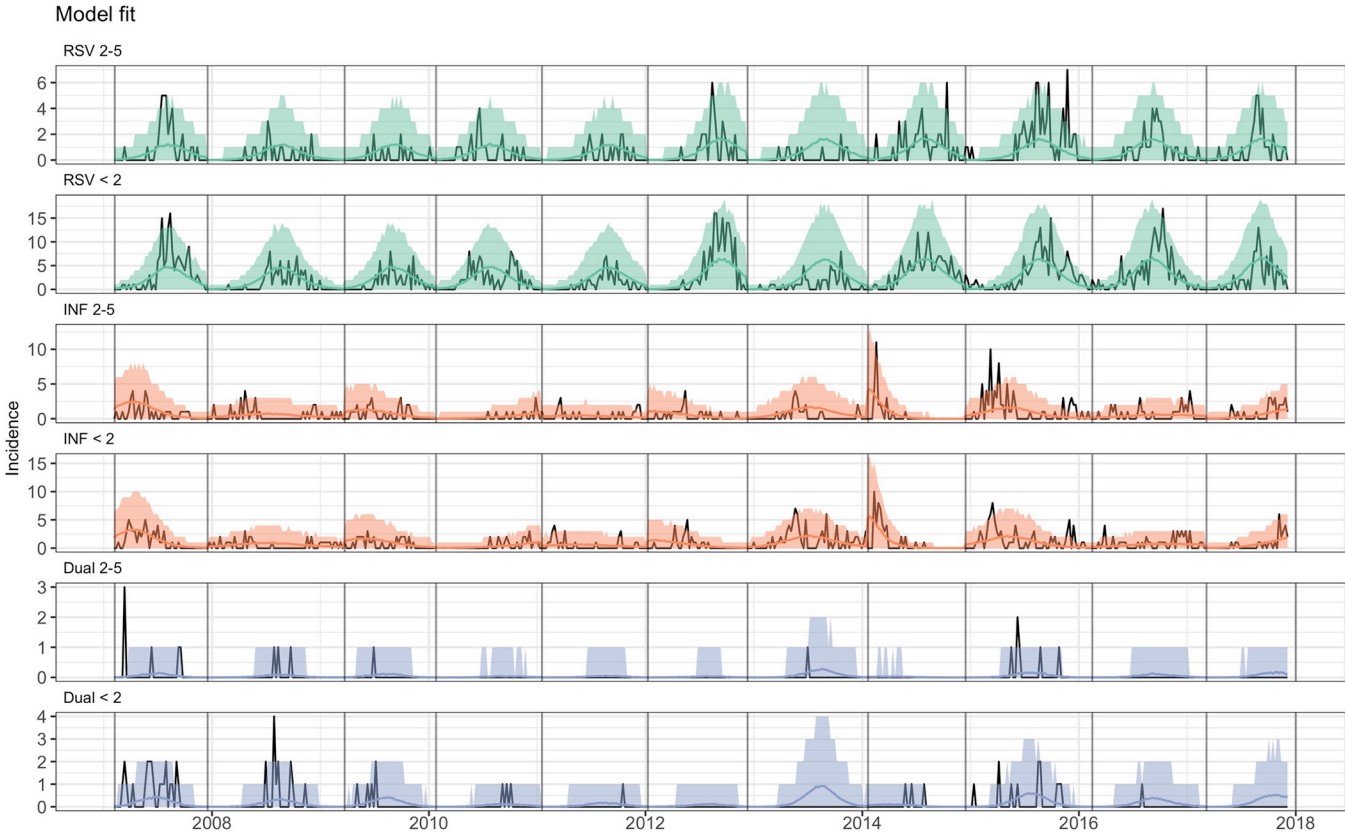

**Fig 2. Model Fit: Black lines are the data, coloured lines are the 95% CrI posterior predictive interval.** Panels show the fit by age group and Virus.

## Sensitivity

As a sensitivity analysis, we assumed that co-infection will not modulate the propensity to be admitted to the local hospital with ARI symptoms (see section 9 in S1 Text), which pushed the posterior to the no competition mode. Further, we reran the model with a prior for strong competition through cross-protection, which pushed the posterior to the moderate interaction mode (see section 10 in S1 Text).

## Discussion

We use data from more than 10 years of enhanced surveillance in Nha Trang, Vietnam to estimate the interaction of influenza and RSV epidemiology. We find that the observed data is consistent with infection reducing heterologous acquisition either by 41% (95%CrI 36% - 54%) for 10.0 days (95%CrI 7.1–12.8 days) after infection or hardly at all. We estimate that influenza-RSV co-infection increases the propensity of an infection to be reported through the ARI hospital surveillance by between 2.5 and 18.4 times. We go on to show that control of one virus in this setting may have little impact on the circulation of the other but can have an added benefit in reducing hospitalisations with co-infections.

A key strength of this dataset is the inclusion of cases infected with both influenza and RSV. Surprisingly though, some dual infections are reported at times when the two viruses seldom found in the hospital setting individually which may be a result of stochastic effect owing to the low number of observed dual infections and has limited the strength of inference from them. While many papers reporting co-infections do not include timings of the co-infections

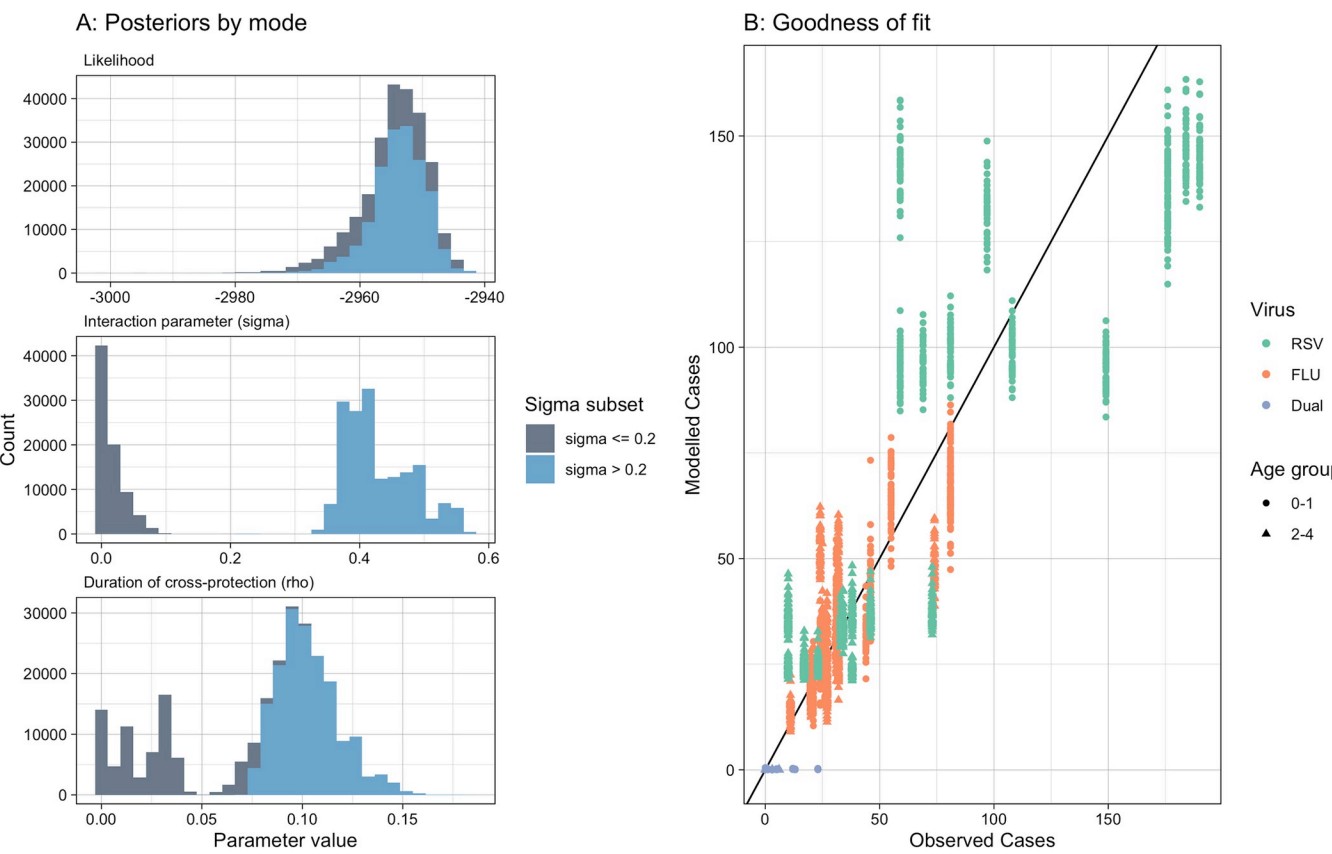

**Fig 3. Posterior estimates for parameters sigma and rho, and the corresponding likelihood values.** Colour is split by sigma value of 0.2. B) Goodness of Fit: Observed cases by season against Modelled cases by season by virus and age group. The black line indicates the same value.

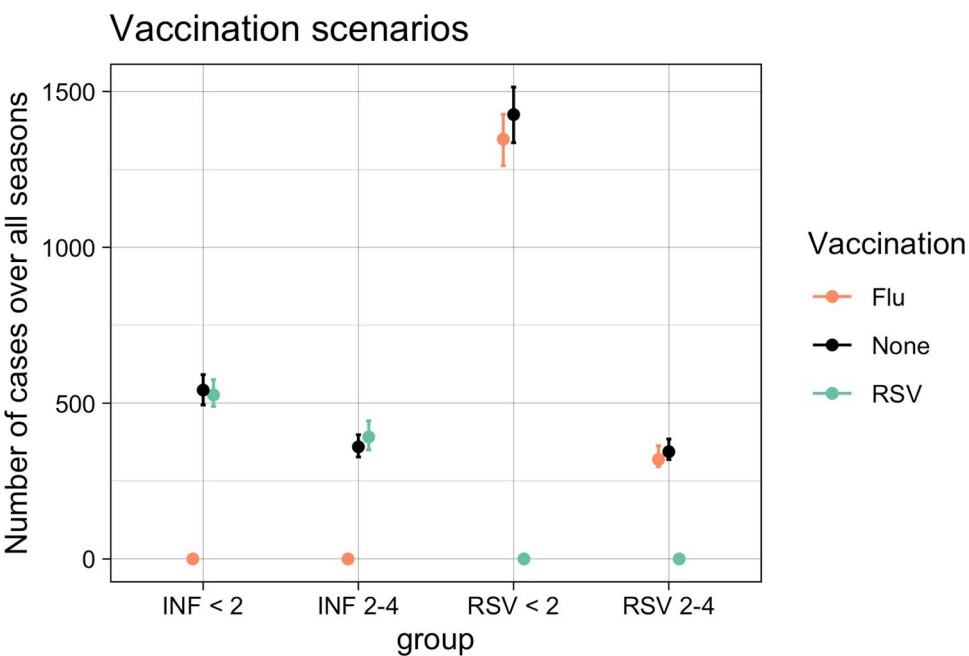

**Fig 4. Vaccination scenarios: Number of cases modelled over all seasons, with different vaccination assumptions.** Dots represent the median and lines the 95% CrI.

[37–39], these off-peak co-infections were not observed in Texas [40]. In addition, in Nha Trang influenza circulates continuously at low prevalence with small epidemics, which would result in constant low-level cross-protection, rather than a short-term more concentrated interaction after a large epidemic, such as in the UK [41]. This low-level cross-protection could have been absorbed into the transmission rate of RSV, explaining the low estimate of the RSV $R_0$ (1.24 compared to a range of 1.2–9.1 in published papers [26]). These location-specific features will need to be considered when generalising the findings. However, in general, the setting of this study is a strength of the paper, as many of the countries where respiratory viruses circulate at low levels year-round are the same countries with limited influenza vaccine uptake. This study also has a high participation rate, however we do not have information on participants who declined to take part in the study, and this may bias the results.

Much of the evidence for cross-protection is on an individual, biological level [5,6,20]. However the impact of this individual level cross-protection on population level has been unclear, due to relatively small infection prevalence at any point in time (we estimate the seasonal attack rate for influenza as 0.011–0.15, and that peak prevalence never exceeded 0.8%), and thus a low propensity for co-infection with RSV. This may be exaggerated by clustering factors such as household transmission, reducing the opportunity for cross-infection. As an example in Kilifi, Kenya, household transmission of RSV contributed about 50% of all RSV transmission in households with young children [42]. Our model assumes a well-mixed population, so does not account for any population clustering beyond the age-specific contact matrices. In addition, we assume that risk of infection is age-dependent, but otherwise homogeneous. However, increased risk of influenza infection may be correlated with increased risk of RSV infection, due to demographic factors such as poor hygiene and household clustering. This may overestimate the effect of dual infections on reporting.

In our model we assumed the cross-protection between the two viruses to be bi-directional, having the same impact irrelevant of which virus caused the first infection. This assumption is based on the mechanism of cross-protection being the activation of the innate immune system into a general antiviral state. However, in the study site the influenza epidemic occurs before the RSV epidemic, therefore our estimate of the strength of cross-protection on susceptibility to the second virus is mainly an estimate of the impact of influenza on RSV and does not necessarily capture any dynamics in the other direction. In addition, our model is not able to capture delays in the timing of cross-protection, which could potentially occur when considering other mechanisms of cross-protection.

Evidence of cross-protection between influenza and RSV also comes from shifts in epidemic peaks, particularly after the 2009 influenza pandemic [10–13]. However these studies are observational, and cannot test mechanisms. As the SARS-CoV-2 pandemic has demonstrated, behavioural responses can have huge impacts on viral circulation, with many geographies seeing shifts in epidemic peaks for usually consistent viruses, such as RSV [43,44], due to limitations on social contacts. Fear generated from high infection rates can also drastically alter individuals behaviour [45], even without wide-spread implementations of restrictions.

Our model does not take into account different subtypes of influenza or RSV, due to the added complexity, additional parameters required and the lack of subtype specific data. We therefore assume that any cross-protection between influenza and RSV does not vary by subtype. We account for different immunity levels to circulating influenza subtypes by fitting a susceptibility parameter at the start of each season. This is necessary because we fit to each season, rather than including immunity waning and fitting over the time period combined. In addition, the dual infections appear to cluster in certain years, and a different explanation of this could be interaction between different viral subtypes each year. While the start weeks of our season are fixed manually, we account for any impacts of this by fitting the proportion

infected at the start of each season for each virus. While most of the posterior estimates are reasonable, the reporting rate for influenza infections is high, between 13 and 21%, compared to estimated 12% from data in Yoshida *et al.* (2013). However, many milder cases (including outpatients) are included in the reports as they may seek healthcare at the hospital, thereby increasing the expected reporting rate in this context. The posterior for the detection rate of influenza is one of the few parameters that, like the strength of cross-protection parameter, is bimodal. However these were not greatly different, with the medians of the two priors only differing by 3.6% and therefore not substantially different to help us distinguish between the two modes. Overall therefore, our model estimates fit the data well, as well as known aspects of influenza and RSV transmission, such as high influenza attack rates in children [46,47], and higher RSV severity in the youngest children [48].

## Conclusions

In summary, we use a novel modelling framework to interrogate a unique case time-series for single and dual infection from Nha Trang, Vietnam. We find that influenza and RSV co-infection substantially increases hospitalisation rates in children. In addition we show that the data supports either no or moderate individual-level cross protection against infection but either way with relatively little population level impact. This alleviates some concerns of heterologous effects of RSV or influenza vaccination, however, particularly in settings with more pronounced and overlapping RSV and influenza seasons the impact of vaccination on the other pathogen's epidemiology may be more noticeable.

## Supporting information

**S1 Text. Further technical details and sensitivity analyses.**
(DOCX)

## Author Contributions

**Conceptualization:** Naomi R. Waterlow, Rosalind M. Eggo, Stefan Flasche.

**Data curation:** Michiko Toizumi, Hien-Anh Thi Nguyen, Lay Myint-Yoshida.

**Formal analysis:** Naomi R. Waterlow.

**Investigation:** Naomi R. Waterlow, Lay Myint-Yoshida.

**Methodology:** Naomi R. Waterlow, Edwin van Leeuwen, Rosalind M. Eggo, Stefan Flasche.

**Project administration:** Naomi R. Waterlow.

**Resources:** Lay Myint-Yoshida.

**Software:** Naomi R. Waterlow.

**Supervision:** Rosalind M. Eggo, Stefan Flasche.

**Validation:** Naomi R. Waterlow.

**Visualization:** Naomi R. Waterlow, Rosalind M. Eggo, Stefan Flasche.

**Writing – original draft:** Naomi R. Waterlow.

**Writing – review & editing:** Naomi R. Waterlow, Michiko Toizumi, Edwin van Leeuwen, Lay Myint-Yoshida, Rosalind M. Eggo, Stefan Flasche.

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
