## [Decision Letter · Decision Letter 0]

18 Jan 2022

Dear Mrs Waterlow,

Thank you very much for submitting your manuscript "Evidence for Influenza and RSV interaction from 10 years of enhanced surveillance in Nha Trang, Vietnam, a modelling study." for consideration at PLOS Computational Biology.

As with all papers reviewed by the journal, your manuscript was reviewed by members of the editorial board and by several independent reviewers. In light of the reviews (below this email), we would like to invite the resubmission of a significantly-revised version that takes into account the reviewers' comments.

The reviewers raise important questions and concerns about the model structure and complexity and the description of the model, which need to be addressed.

We cannot make any decision about publication until we have seen the revised manuscript and your response to the reviewers' comments. Your revised manuscript is also likely to be sent to reviewers for further evaluation.

Sincerely,

Daniel M Weinberger

Guest Editor

PLOS Computational Biology

Virginia Pitzer

Deputy Editor-in-Chief

PLOS Computational Biology

The reviewers are generally enthusiastic about the analysis but raise important questions and concerns about the model structure and complexity, which need to be addressed.

Reviewer's Responses to Questions

**Comments to the Authors:**

Reviewer #1: Overall, this is a well-done and clearly-written analysis of interactions between Influenza and RSV using detailed surveillance data from Vietnam. The authors developed an age-stratified transmission model for RSV and influenza, which was fitted to children aged <5 years with either one or two respiratory viruses infections to determine the strength of cross-protection and the effects of co-infections on disease severity. The authors concluded that co-infections increase the probability of being hospitalized and had a bimodal conclusion on cross-protection. The methods employed are well-justified and clearly articulated, and the conclusions appear to be broadly consistent with the results presented. However, I do have a few major comments regarding the methodology.

[1] The authors fitted 15 parameters for each season individually (line 159 and table 1). The maximum time period for each season in the model is 66 weeks and it seems that RSV infections are mostly concentrated between November and May (line 172-line 173 and Figure 1C), which are about 26 weeks. I am concerned that the sparsity of the data may cause issues of parameter identifiability. I find it hard to be convinced that the whole model converged well giving that several parameters showed bimodal posterior distributions. I would recommend the authors use the entire 10 year period with yearly varying parameters of influenza and use weakly informative priors based on published literature, especially for those 0-inf priors.

[2] In the Supplement, Model equations section, the outflow from line 68 -(1-sigma)*lambda_INFi*PS_i does not equal to the inflow in line 72 lambda*PS_i. Also, if the epsilon means introduction rate from external sources, it should not show up as outflows from the susceptible population. It would also help readers to understand the equations if the authors can specify the meaning of each compartment like what is done in main text 138-139 and the captions of Figure 2C.

Minor comments:

Introduction

[3] It may be better to clarify what is the “interaction” that this paper refers to. In lines 88 and 89, “to estimate the strength of cross-protection and the effects of co-infections on disease severity” may be more precise as it corresponds to the conclusions.

Methods

[4] Line 98: Do you have information on the 6% that were not enrolled in the study? Are their demographic background (remote living? who may have less chance of interactions with those living in the city) and contact patterns (such as household size) the same as those enrolled in the study?

[5] Line 100: Can you provide the sensitivity and specificity of the multiplex-PCR test?

[6] Line 198: Please provide information on how many interactions were left for the parameter estimates.

Reviewer #2: PCOMPBIOL-D-21-01994

The authors aim to better quantify the interaction between influenza virus and RSV, using data on pediatric infection from Nha Trang in Vietnam, and using a transmission modeling approach.

I appreciate the challenging nature of this type of study. But in this current form, I find the study lacking in several ways, and I have tried to describe these below. Perhaps some of these shortcomings can be addressed, in which case, it could be a worthwhile contribution to this field.

1. Important parts of the transmission model (and data) are not presented or adequately described.

It is unclear what assumptions the authors make about the dynamics of and transmission from individuals >5 years? It is stated in the methods that older age groups are modeled (which is good), but the data used to calibrate these, and the results (ie, incidence of flu and RSV from model simulations) are lacking.

Given that >5 population will be substantially larger, and may play an overriding role in driving the dynamics of pediatric infections, it should be described with more clarity and supported with data.

Similarly, it is unclear if and how seasonality is modeled. Given seasonal nature of transmission, especially of RSV, I think it would be important to carefully include this.

2. The model for interaction between influenza and RSV, is not conceptualized or described in detail.

In the background, authors describe that 1. increased severity of co-infections and 2. cross protection between two viral infections as the two pathways of interactions that the authors are interested in estimating.

However, it is unclear how is severity of co-infections are modeled; and what data are used to measure severe forms of co-infections.

For example, cross protection is modeled in a specific way, and it is unclear to me that the assumptions are justified. For example, it is assumed to be symmetric -- that protection from an infection of influenza to RSV would be the same in the opposite direction?

The evidence that authors cite in the background are mostly protection from prior influenza infection on RSV. At the very least, the modeling framework should be flexible to allow for protective interaction in one direction.

The fact that they find bimodality in their results may also be pointing to this.

Relatedly, while the duration of the cross protection is explored, the onset of the cross-protection is assumed to begin instantaneously. It would be importantly explore the timing, especially given that some of the studies they site report protection after several weeks.

3. I find it hard to make sense of the findings -- 41% cross-protection for 10 days or nothing. I can make sense of a finding where the data and the model are not able to find statistically significant evidence of interaction -- that no interaction and some interactions are equally likely.

But from what the authors are presenting, it appears to argue for two very distinct, yet very clearly defined interactions, which seems to be contradictory. I think this aspect needs to be explored in greater detail.

A few options may be:

(i) exploring more flexible forms of interaction (eg, allowing for asymmetry or delay in onset, as I have argued in 2)

(ii) showing simulations of two explanations -- whether one model captures certain features of the data better.

(iii) showing profile likelihoods for rho, to clearly show how evidence for the estimates compare.

Minor points/comments:

Abstract:

Line 24: Unclear what "heterologous ecological changes" means. Suggest simplifying the language.

Lines 31-33: The statement seems confusing or contradictory. Is this meant to suggest that you data cannot distinguish between

Lines 33-35: I do not understand what the two respective scenarios with 7.2 and 16.6 fold increases are -- perhaps something is missing?

Background:

One thing to point out in describing the two potential types of interactions, increased severity of co-infections and protective effect of influenza on RSV, is also the timing of those.

Clarifying in these studies the evidence for concurrent infections vs consecutive ones, would be helpful to the readers.

Perhaps useful to stipulate the specific types of cross-protection you are describing, particularly those in which two viral agents are not related antigenically.

Methods

It would useful to know the population denominators for Nha Trang.

It is unclear how seasonality is modeled, if it is at all?

Figure 1: Age groups do not include 4-5 years?

Figure 2: Part of this figure is illegible, especially panel C, model diagram, both due to smal text size and the quality of the figure.

Reviewer #3: Overall I think the authors present a strong, well-written, and interesting paper that will help improve our understanding of the interactions between various respiratory pathogens, and the implications for public health. I should make clear that I am not an expert in mathematical modeling, though what the authors describe having done seems reasonable to me. I do have a few minor comments/suggestions for the authors that I think will help improve the overall clarity and readability of the manuscript.

Comments:

Pg 7 (lines 62-63): This study was only in adults. The evidence in children has been far more mixed. Edit: I see you addressed this later on!

Pgs 10-11 (lines 144-145) and pg 11 (lines 162-163): Upon first reading these sentences they appear to contradict each other. After looking at it a bit more closely it becomes clear that they do not, but I would recommend revising these sentences to make this more apparent. Your language in the discussion (pg 22, lines: 343-345) was particularly helpful in understanding the reasoning behind your approach, but addressing this earlier in the methods of the paper would be helpful.

Pg 15 (lines: 209-212): The reasoning behind the second sensitivity analysis is a bit confusing as it is currently written. I would suggest the authors revise these sentences a bit to make this clearer.

Pg 19 figure 2A: It appears that the dual infections tended to cluster in certain years which might indicate the importance of subtype specific interactions. While I appreciate the substantial complexity that influenza and RSV subtypes would add to the model, if you have data on the influenza and/or RSV subtypes that tended to predominate in years where dual infections were more/less frequently observed this would be quite helpful to include in the descriptive analysis.

Pgs 20-21 (lines: 310-313): The setting of this analysis I think is a strength as many of the countries where respiratory viruses circulate at low levels year-round are the same countries with limited influenza vaccine uptake.

**Have the authors made all data and (if applicable) computational code underlying the findings in their manuscript fully available?**

Reviewer #1: Yes

Reviewer #2: **No: **Please refer to general comments

Reviewer #3: Yes

PLOS authors have the option to publish the peer review history of their article (what does this mean?). If published, this will include your full peer review and any attached files.

Reviewer #2: No

Reviewer #3: No
---

## [Decision Letter · Decision Letter 1]

20 May 2022

Dear Mrs Waterlow,

We are pleased to inform you that your manuscript 'Evidence for Influenza and RSV interaction from 10 years of enhanced surveillance in Nha Trang, Vietnam, a modelling study.' has been provisionally accepted for publication in PLOS Computational Biology.

You may also wish to address the small (optional) comment from the reviewer below.

Best regards,

Daniel M Weinberger

Guest Editor

PLOS Computational Biology

Virginia Pitzer

Deputy Editor-in-Chief

PLOS Computational Biology

Thanks for your re-submission and apologies for the long review--it was difficult to wrangle the reviewers to have a look at it. I am recommending acceptance; Reviewer 1 does have 1 comment that could be addressed if you would like to address it with a minor revision. Comment from Reviewer 1: "One thing that can be improved is the discussion of simulation as shown in reviewer 2’s comment. The author can discuss how well their model structure is able to capture the parameter space based on their previously published simulation study entitled Competition between RSV and influenza: Limits of modelling inference from surveillance data."

Reviewer's Responses to Questions

**Comments to the Authors:**

Reviewer #1: The authors addressed all the comments pretty well. I do not have any further comments except for one, which is also optional.

One thing that can be improved is the discussion of simulation as shown in reviewer 2’s comment. The author can discuss how well their model structure is able to capture the parameter space based on their previously published simulation study entitled Competition between RSV and influenza: Limits of modelling inference from surveillance data.

**Have the authors made all data and (if applicable) computational code underlying the findings in their manuscript fully available?**

Reviewer #1: Yes

PLOS authors have the option to publish the peer review history of their article (what does this mean?). If published, this will include your full peer review and any attached files.

Reviewer #1: **Yes: **Zhe Zheng

---

## [Editor Report · Acceptance letter]

21 Jun 2022

PCOMPBIOL-D-21-01994R1 

Evidence for Influenza and RSV interaction from 10 years of enhanced surveillance in Nha Trang, Vietnam, a modelling study.

Dear Dr Waterlow,

I am pleased to inform you that your manuscript has been formally accepted for publication in PLOS Computational Biology. Your manuscript is now with our production department and you will be notified of the publication date in due course.

With kind regards,

Marianna Bach
